# Rapid Acquisition, Management, and Analysis of Spatial Maize (*Zea mays* L.) Phenological Data—Towards 'Big Data' for Agronomy Transformation in Africa

**Henri E. Z. Tonnang** [1,2,3,*], **Tesfaye Balemi** [4], **Kenneth F. Masuki** [1,5], **Ibrahim Mohammed** [2,6], **Julius Adewopo** [2] ⬤, **Adnan A. Adnan** [6], **Bester Tawona Mudereri** [3] ⬤, **Bernard Vanlauwe** [2] and **Peter Craufurd** [1] ⬤

[1] International Maize and Wheat Improvement Center (CIMMYT), off UN Avenue, Gigiri, ICRAF House, Nairobi P.O. Box 1041-0062, Kenya; kennymasuki@gmail.com (K.F.M.); p.craufurd@cgiar.org (P.C.)
[2] International Institute of Tropical Agriculture (IITA), PMB 5320, Oyo Road, Ibadan 200001, Nigeria; Ibmohammed.agr@buk.edu.ng (I.M.); J.Adewopo@cgiar.org (J.A.); B.Vanlauwe@cgiar.org (B.V.)
[3] International Centre of Insect Physiology and Ecology (icipe), Nairobi P.O. Box 30772-00100, Kenya; bmudereri@icipe.org
[4] Ethiopian Institute of Agricultural Research, Addis Ababa 2003, Ethiopia; tesfayeb2005@yahoo.co.uk
[5] Global Water Partnership Tanzania (GWPTZ), Mikocheni B, 196 Rose Garden Road, Dar Es Salaam P.O. Box 32334, Tanzania
[6] Agronomy Department, Bayero University, Kano PMB 3011, Nigeria; aaadnan.agr@buk.edu.ng
* Correspondence: htonnang@icipe.org

**Abstract:** Mobile smartphones, open-source set tools, and mobile applications have provided vast opportunities for timely, accurate, and seamless data collection, aggregation, storage, and analysis of agricultural data in sub-Saharan Africa (SSA). In this paper, we advanced and demonstrated the practical use and application of a mobile smartphone-based tool, i.e., the Open Data Kit (ODK), to assemble and keep track of real-time maize (*Zea mays* L.) phenological data in three SSA countries. Farmers, extension agents, researchers, and other stakeholders were enlisted to participate in an initiative to demonstrate the applicability of mobile smartphone-based apps and open-source servers for rapid data collection and management. A pre-installed maize phenology data application based on the ODK architecture was provided to the participants (n = 75) for maize data collection and management over the maize growing season period in 2015–2017. The application structure was custom designed based on maize developmental stages such as planting date, date of emergence, date of first flowering, anthesis, grain filling, and maturity. Results showed that in Ethiopia, early maturing varieties took 105 days from sowing to maturity in low altitudes, whereas late-maturing varieties took up to 190 days to complete developmental stages in high altitude areas. In Tanzania, a similar trend was observed, whereas in Nigeria, most existing varieties took an average of 100 days to complete their developmental stages. Furthermore, the data showed that the durations from sowing to emergence, emergence to flowering, flowering to maturity were mainly dependent on temperature. The values of growing degree for each phase of development obtained from different planting dates were almost constant for each maize variety, which showed that temperature and planting time are the main elements affecting the rate of maize development. The data aggregation approach using the ODK and on-farm personnel improved efficiency and convenience in data collection and visualization. Our study demonstrates that this system can be used in crop management and research on many spatial scales, i.e., local, regional, and continental, with relatively high data collation accuracy.

**Keywords:** applications (apps); growing degree days; maize; Open Data Kit (ODK); server; smartphone

## 1. Introduction

The implementation and adoption of agronomic activities aimed at improving farm productivity in developing countries are gaining momentum [1]. Lately, many agronomists have explored ways to integrate science and technology with farms and farming systems at different spatial and temporal scales [2–5]. Precision agriculture requires spatially explicit information of on-farm data because of landscape variations such as soil properties, climatic conditions, and crop as well as nutrient management, cultural practices, and climatic factors [6,7]. Therefore, modern day agronomy emphasizes the need to enumerate the spatiotemporal variations in crop and soil conditions at very high spatial resolutions [4,8]. This drives the need for agronomic research to evaluate the impact of such variations on crop yields and develop appropriate tools and recommendations to enable improvement in productivity and farm management techniques. However, the inadequacy of data at desirable spatial and temporal resolutions has greatly constrained the prospects of developing and applying site-specific tools for the optimization of crop yields.

Conducting an exhaustive ground-based collection of data on biophysical factors that influence crop yields is often cost-prohibitive. However, the advancement in Information, Communication and Technology (ICT), particularly wireless, portable computer gadgets, personal digital assistants (PDAs), tablets, and mobile smartphones, has provided efficient and cost-effective methods of data collection [9–11], thus reducing cost and increasing the speed and quality of data collection. This has been further enhanced by the advent, rapid growth, and widespread coverage of mobile communication technology. The World Bank reported that approximately three-quarters of the world population had access to mobile smartphones [12] and that approximately 80% of the active mobile phone subscriptions were in developing countries. This proliferation of mobile smartphones provides an opportunity for their extensive use in data collection. Although several studies have explored the applicability of mobile smartphones and associated open source applications [13,14] to enhance data collection in health [15] and social settings [16], limited mobile smartphone-based capabilities have been developed to support smallholder farmers in sub-Saharan Africa (SSA). For instance, in Western Tanzania, a report by Dillon [17] mentioned the use of a short message service (SMS)-based data collection method for monitoring climatic and agronomic inputs. However, using the SMS platform limits the seamless aggregation of data originating from different sources.

Nonetheless, mobile smartphones have benefited farmers in different countries to access information such as transport to markets, agricultural techniques, and market prices [18]. Thus, the rapid advancement that has been afforded by the use of mobile smartphone technologies has improved agricultural operations at the farm level and in the interaction and engagement with farmers in Africa [19]. Moreover, the increased ease of utilizing web-based technologies and the recent advent of cloud-based storage for big data management is an important driver that provides new opportunities for developing data-rich decision support systems across larger agricultural landscapes [19].

In this study, we integrated the acquisition of geospatially referenced data into conventional agronomy to achieve an increased understanding of within-farm and between-farm variability in maize (*Zea mays* L.) phenology using the mobile smartphone and the Open Data Kit (ODK) tools. We generated a large pool of data to enhance decision support systems analytics within core maize producing areas in Ethiopia, Nigeria, and Tanzania. The data collated from the farmers were then used to estimate the length of the growing season of different maize varieties in different locations that experience different climatic conditions (in the case of Ethiopia and Tanzania), and we estimated the key parameters that influence the development of the maize plants. Maize is critical in the African continent as it secures the food and nutrient needs of around 1.2 billion people [20,21]. As the responses of maize plant to environmental conditions vary among varieties, a correct and timely assessment of the developmental rate can provide a guide in estimating the timing and duration of critical periods

for plant growth, which in turn affects both the quantity and quality of grain yield [22]. Furthermore, to improve maize production in Africa, the quality of the maize data should be easily collected and visualized in the minimum time possible [23]. On this basis, the ODK was tested through this study to speed up data collection and ensure the availability of quality and validated maize development information for decision-making.

Therefore, this study was conducted with the following objectives: (1) to develop and test a cost- and time-effective near-real-time maize growth tracking application using ODK and associated tools; (2) to evaluate the efficacy of the developed prototype for data collection, entry, and analysis of maize phenology data; (3) to improve the aggregation of field data and real-time visualization of data using the different components of ICT.

## 2. Materials and Methods

### 2.1. Study Areas

This study was carried out in core maize growing areas of Ethiopia, Nigeria, and Tanzania (Figure 1). These areas are defined as having diverse temperatures, especially in Ethiopia and Tanzania. The study sites covered 5 locations (Ethiopia), 5 locations (Nigeria), and 10 locations (Tanzania) within the major maize production pilot areas in the individual countries. The maize target region was defined as the area where >50% of the maize production in the respective countries is derived based on the Harvest Choice database [24], which draws data from other sources such as the International Food Policy Research Institute (IFPRI) and FAO databases.

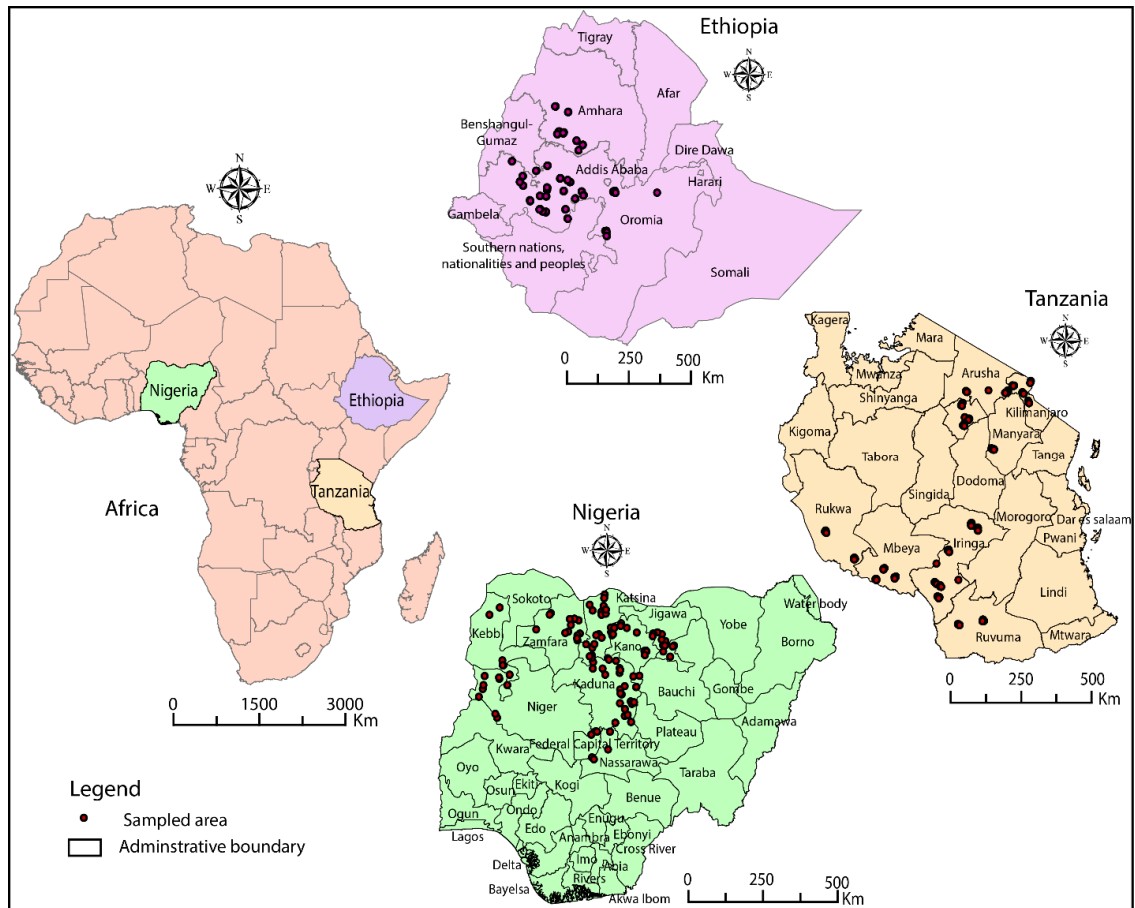

**Figure 1.** The location of the three countries in Africa and the relative locations of the maize trial sites used in this study.

### 2.2. Experimental Design

The trials were arranged in completely randomized block design (CRBD), with each variety planted on 3 planting dates and with 3 replicates at each site. The size of each plot was 2.25 m × 5.5 m. Plant spacing of 0.75 m between rows and 0.25 m between plants within a row was used, making 22 plants per row, with an approximation of 6 plants per square meter. Main observations recorded include the sowing date, days to emergence, days to flowering, and days to physiological maturity. Emergence was measured by counting the number of maize plants daily and the emergence date was considered when 50% of the plants in the plot emerged. Ten (10) to 15 days after the emergence of all plants, 10 plants were randomly selected within the middle (4 rows × 1.5 m) in each plot and tagged with colored material. The selected 10 plants were used to record the vegetative phase (VP) and the reproductive phase (RP). The flowering date was recorded when 60% (6 plants) out of the tagged 10 plants had flowered. Similarly, days to physiological maturity was recorded when 60% of the tagged 10 plants had reached maturity.

### 2.3. Training Program of Data Collectors

Before the start of the growing season, a 3-day training session was conducted in each country to acquaint data collectors (n = 25 in each country) with the use of the mobile smartphones and apps for successful data collection within the target region. Demonstration sessions were conducted to test the steps of collecting information for correctness and clarity. Standard protocols and operating procedures were developed for sample collection. Pre-formatted blank forms were loaded from the server into the mobile smartphone devices of the data collectors and the collectors were guided to fill, edit, save, and transfer each completed form to the online ONA (https://ona.io/) server. After the training, a field pre-test of the collection protocol was conducted before the actual field collection.

### 2.4. Data Collection and Application Workflow

The features for collection, transfer, and archiving of geospatially referenced data of the ODK app were used. This is a multi-functional application (app) platform designed to collect, aggregate, and analyze survey data [25,26]. The steps of gathering the information were composed in the 'ODK Build' module and were subsequently deployed to the 'ODK Collect' module in the Android device. This module also serves as the interface between forms and the server. The 'ODK Aggregate' module is a server from which blank forms are accessed by 'ODK Collect' and where data from completed forms are stored. Additionally, 'ODK Aggregate' allows for data access and visualization arranged as charts, maps, and images [25,26]. ODK renders complex application logic and supports the manipulation of data types that include text, location, images, audio, video, and barcodes. The ODK format was developed based on XLSForm standards (Figure 2) which provided appropriate formatting syntax in the human-readable language of Microsoft Excel (XLS format) before being translated to XML schemas in the mobile smartphones for data collection and storage [27].

ONA cloud server (ONA: https://ona.io/) was used to publish and store blank forms and served as a temporary repository for the overall collected information. For correct customization of the data collection steps regarding the ODK platform, an account was created within the ONA server through which the blank forms were published, managed and from where collected information was submitted and accessed. The ONA account provides a secure connection and allows an administrator to define the access rights to the forms and the information collected from the field. After cleaning and removal of sensitive information, the data were transferred to CIMMYT's Data Verse repositories and used in this analysis.

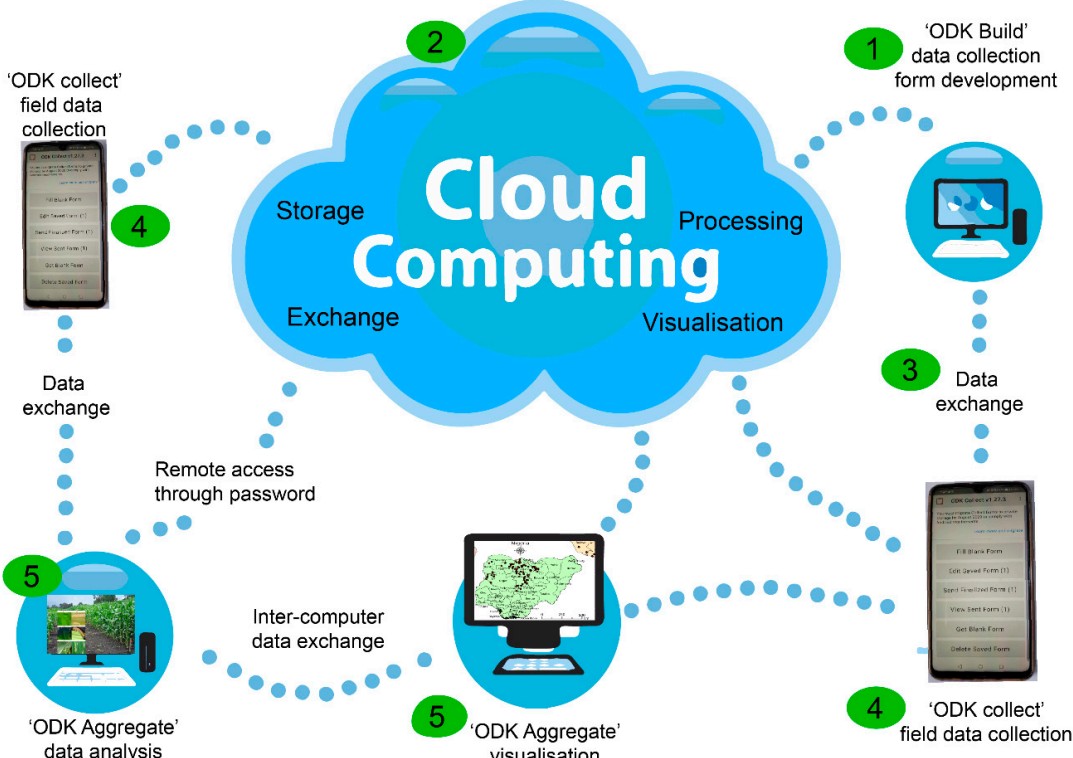

**Figure 2.** Open data kit (ODK) data collection workflow and the mobile smartphone interface for data collection through ODK Collect.

*2.5. Maize Plant Phenology*

The government recommended and the most cultivated varieties in the different countries of the study were used to test the ODK data collection framework. The varieties that were used correspond to the variety numbers 20, 55, and 42 (Tables S1–S3) hybrid and open-pollinated varieties (OPV) of maize. Data on maize phenology were recorded under a large temperature gradient to estimate how the duration of each phenological phase responds to diverse growing conditions. The selection of experimental fields was guided by the presence of weather stations in the vicinity. The maize phenology stages that were used in this study are shown in Figure 3. We considered the two maize phenology phases: the vegetative phase (VP) and the reproductive phase (RP). The VP starts at the emergence stage (VE), goes through the $n$th leaf stage (Vn), and ends with the tasseling stage (VT). In VE, most of the growth occurs beneath the soil surface [28]. Depending on the maize variety and the environment, V5, V9, and V15 may occur between 14 and 21, 28 and 35, and 56 and 63 days after emergence, respectively. The succeeding 6 stages are part of the RP: silking, blister, milk, dough, dent, and maturity [28]. However, care was taken to establish these stages because the speed of the loss of greenness and maize vigor depends on the variety and the growing conditions.

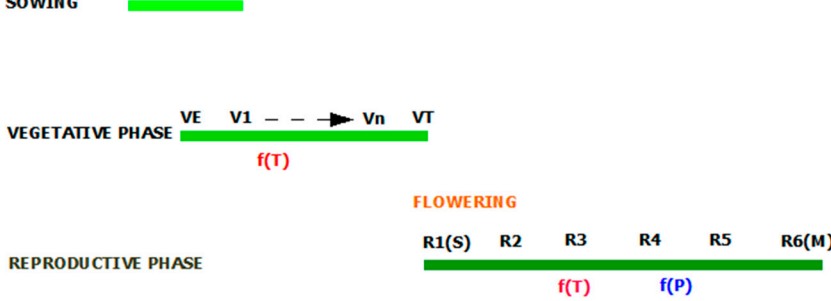

**Figure 3.** The maize phenology stages considered in this study.

### 2.6. The Accumulated Temperature during Growing Seasons

The overall developmental rate per developmental phase for the maize plant is estimated by accumulating the daily development rate values [29]. Practically, the duration to complete the phase is the inverse to estimate the development rate, and the growing degree (GDD) days was given by the following mathematical expression [30,31]:

$$GDD = \sum_{0}^{n} \left[ \frac{T_{max} + T_{min}}{2} \right] - T_b \tag{1}$$

where $T_{max}$ $T_{min}$ and $T_b$ are the daily maximum, minimum, and base temperatures during the growing season, respectively. $T_b$ was assumed to be 8, 10, and 10 °C for Ethiopia, Nigeria, and Tanzania, respectively. The upper limit above which temperature becomes stressful to maize development in all countries was set at 30 °C. Estimates of the accumulated temperature during growing seasons were restricted to only Ethiopia and Nigeria as we were not able to obtain satisfactory climate data for Tanzania.

## 3. Results

### 3.1. Capabilities of the ODK Platform in Agriculture Research

The use of mobile smartphones equipped with cameras allowed the collection of plant leaf reference images on-farm (Figure 4). The overall process of data collection provided a prerequisite for quality checks, which were performed in real-time. This allowed for the detection of data collection flaws for appropriate correction and cleaning within the data collection period. In this study, there was no reported data loss, nor was there any incidence of hardware/software failure.

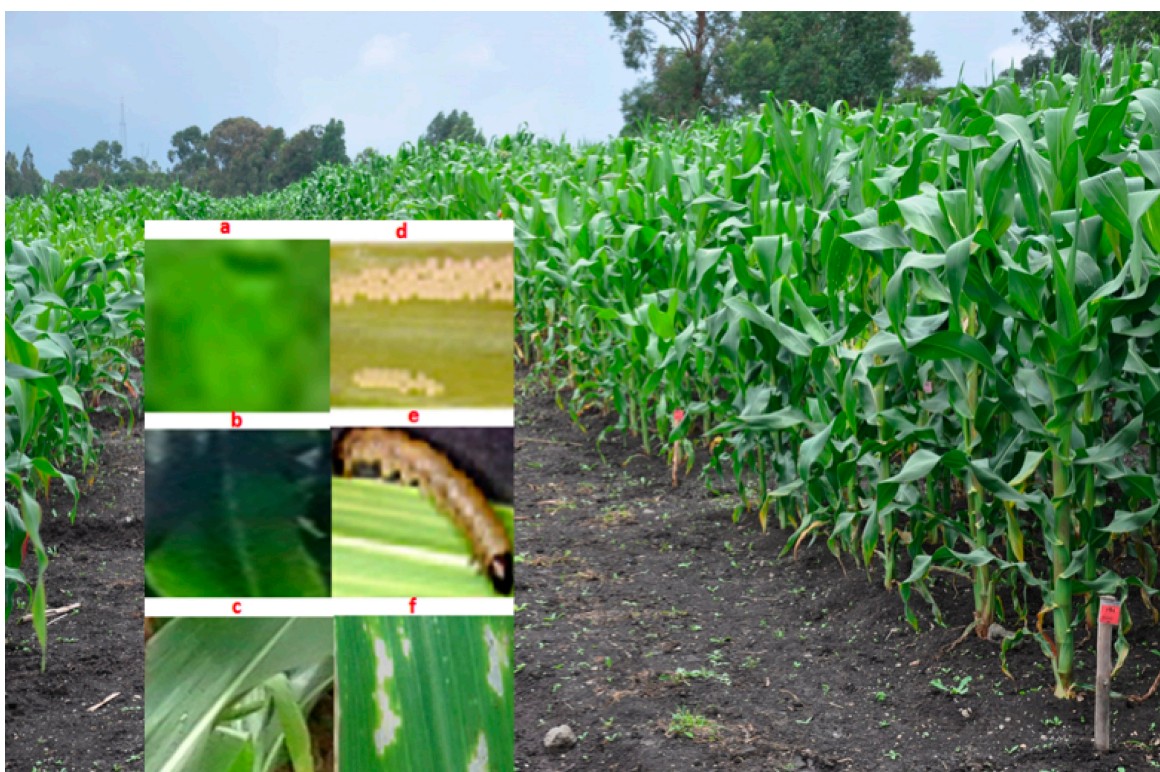

**Figure 4.** Example images collected using the ODK of a trial plot, (**a–c**) close images of leaves, (**d**) leaf of maize with insect egg, (**e**) insect larvae, and (**f**) signs of damage to the maize by disease.

The use of mobile smartphones, with Internet and cloud server connection, enabled easy transfer of the collected information to a secure cloud-based repository. These survey tools provided instantaneous means for project managers to track errors and scrutinize the data for any inconsistency that could compromise the project goals. Through the inbuilt graphs and reports in ONA, the project manager in each country could visualize outputs such as data completion count on an hourly and daily basis as well as the average completion time. Moreover, the ONA platform provided online tools that allowed viewing and interacting with the obtained data in different formats (.csv, .xls, and .sav). A practical challenge that was noticed during the data collection was in the fast reduction rate of mobile smartphone battery usage. Locations of farms were often away from a power source and data collectors were sometimes obliged to carry several mobile smartphones and additional power banks to counter this challenge.

### 3.2. Maize Phenology

Classification of maize varieties in Ethiopia and Tanzania differed from the grouping in Nigeria (Tables S1–S3). In the latter country, the emphasis was placed on the type of soil and vegetation (savanna or forest), whereas, in Ethiopia and Tanzania, altitude played an important role in differentiating maize varieties. Three elevation classes (low, medium, and high) are associated with the altitude measurements to characterize the ecology, which is translated to the difference in the duration of the phenology phases of the plant.

Depending on the locations, the difference in the number of days that elapse from sowing to maturity varied with altitude. In Ethiopia, the early maturing varieties took 105 days from sowing to maturity in low altitudes, whereas the late maturing variety took ~190 days to complete developmental stages in high altitude areas. In Tanzania, a comparable trend was found, whereas in Nigeria, most of the existing varieties took around 100 days to complete the developmental stages. Only in middle belt Nigeria, the region of Joss that experiences cold environmental and climate conditions, were varieties allowed to spend more than 100 days from sowing to physiological maturity (Tables 1 and 2).

**Table 1.** Values of ≥ 10 °C accumulated temperature and growing degree day (GDD) requirements during different growth periods with 3 planting dates for 2 maize varieties (PHB30-G19 and Melkasa-4) planted in 3 locations (DEDESSA, UKE, and AMBO) in Ethiopia.

| Locations | Maize Variety Names | Planting Dates | Accumulated Temperature | Growth Periods | | |
| --- | --- | --- | --- | --- | --- | --- |
| | | | | Sowing to Emergence (°C days) | Emergence to Flowering (°C days) | Flowering to Maturity (°C days) |
| DEDESSA | PHB30-G19 (Hybrid) | 23/05/2016 | ≥8 °C GDD | 122 (10) | 898 (35) | 809 (42) |
| | | 30/05/2016 | ≥8 °C GDD | 118 (11) | 901 (39) | 830 (25) |
| | | 06/06/2016 | ≥8 °C GDD | 113 (15) | 894 (42) | 817 (33) |
| | Melkasa-4 (OPV) | 23/05/2016 | ≥8 °C GDD | 122 (10) | 878 (35) | 887 (47) |
| | | 30/05/2016 | ≥8 °C GDD | 118 (11) | 892 (33) | 873 (31) |
| | | 06/06/2016 | ≥8 °C GDD | 113 (15) | 835 (41) | 888 (32) |
| UKE | PHB30-G19 (Hybrid) | 25/05/2016 | ≥8 °C GDD | 115 (10) | 853 (40) | 848 (45) |
| | | 31/05/2016 | ≥8 °C GDD | 112 (10) | 862 (31) | 807 (42) |
| | | 07/06/2015 | ≥8 °C GDD | 112 (9) | 852 (52) | 874 (39) |
| | Melkasa-4 (OPV) | 25/05/2016 | ≥8 °C GDD | 115 (10) | 814 (36) | 875 (28) |
| | | 31/05/2016 | ≥8 °C GDD | 112 (10) | 817 (28) | 834 (39) |
| | | 07/06/2015 | ≥8 °C GDD | 112 (9) | 860 (41) | 880 (35) |
| AMBO | PHB30-G19 (Hybrid) | 28/05/2016 | ≥8 °C GDD | 108 (13) | 846 (33) | 812 (25) |
| | | 07/06/2016 | ≥8 °C GDD | 120 (15) | 890 (29) | 887 (28) |
| | | 15/06/2016 | ≥8 °C GDD | 111 (15) | 884 (31) | 883 (29) |
| | Melkasa-4 (OPV) | 28/05/2016 | ≥8 °C GDD | 117 (13) | 899 (27) | 809 (21) |
| | | 07/06/2016 | ≥8 °C GDD | 110 (15) | 900 (23) | 848 (16) |
| | | 15/06/2016 | ≥8 °C GDD | 109 (15) | 878 (19) | 816 (18) |

**Table 2.** Values of ≥ 10 °C accumulated temperature and growing degree day (GDD) requirements during different growth periods with 2 planting dates for 3 maize varieties (SAMMAZ 32, SAMMAZ 15, and IFE HYBRID 5) planted in 4 locations (Bayero University Kano (BUK), DAMARU, SAMARU, and LERE) in Nigeria.

| Locations | Maize Variety Names | Planting Dates | Accumulated Temperature | Growth Periods | | |
| --- | --- | --- | --- | --- | --- | --- |
| | | | | Sowing to Emergence (°C days) | Emergence to Flowering (°C days) | Flowering to Maturity (°C days) |
| BUK | SAMMAZ 32 (OPV) | 12/03/2016 | ≥10 °C GDD | 71 (7) | 909 (16) | 878 (14) |
| | | 10/06/2016 | ≥10 °C GDD | 69 (5) | 892 (14) | 804 (12) |
| | SAMMAZ 15 (OPV) | 12/03/2016 | ≥10 °C GDD | 92 (8) | 1123 (19) | 984 (17) |
| | | 10/06/2016 | ≥10 °C GDD | 103 (8) | 1007 (18) | 970 (16) |
| | IFE HYBRID 5 (Hybrid) | 12/03/2016 | ≥10 °C GDD | 70 (6) | 829 (10) | 809 (10) |
| | | 10/06/2016 | ≥10 °C GDD | 69 (6) | 807 (9) | 781 (9) |
| DAMBATTA | SAMMAZ 32 (OPV) | 20/03/2016 | ≥10 °C GDD | 68 (9) | 958 (16) | 876 (14) |
| | | 16/06/2016 | ≥10 °C GDD | 71 (9) | 949 (15) | 820 (13) |
| | SAMMAZ 15 (OPV) | 20/03/2016 | ≥10 °C GDD | 101 (13) | 1311 (23) | 1010 (18) |
| | | 16/06/2016 | ≥10 °C GDD | 89 (11) | 1104 (19) | 959 (17) |
| | IFE HYBRID 5 (Hybrid) | 20/03/2016 | ≥10 °C GDD | 68 (9) | 907 (17) | 887 (16) |
| | | 16/06/2016 | ≥10 °C GDD | 71 (9) | 913 (17) | 861 (16) |
| SAMARU | SAMMAZ 32 (OPV) | 16/03/2016 | ≥10 °C GDD | 84 (4) | 875 (14) | 815 (12) |
| | | 20/06/2016 | ≥10 °C GDD | 87 (5) | 826 (9) | 795 (5) |
| | SAMMAZ 15 (OPV) | 16/03/2016 | ≥10 °C GDD | 101 (6) | 1087 (15) | 906 (7) |
| | | 20/06/2016 | ≥10 °C GDD | 105 (8) | 933 (13) | 893 (6) |
| | IFE HYBRID 5 (Hybrid) | 16/03/2016 | ≥10 °C GDD | 84 (4) | 804 (10) | 798 (5) |
| | | 20/06/2016 | ≥10 °C GDD | 70 (4) | 800 (9) | 787 (4) |
| LERE | SAMMAZ 32 (OPV) | 21/03/2016 | ≥10 °C GDD | 84 (6) | 990 (8) | 931 (8) |
| | | 13/06/2016 | ≥10 °C GDD | 87 (7) | 879 (6) | 836 (5) |
| | SAMMAZ 15 (OPV) | 21/03/2016 | ≥10 °C GDD | 100 (9) | 1042 (11) | 895 (7) |
| | | 13/06/2016 | ≥10 °C GDD | 103 (9) | 943 (9) | 873 (8) |
| | IFE HYBRID 5 (Hybrid) | 21/03/2016 | ≥10 °C GDD | 84 (4) | 915 (6) | 814 (8) |
| | | 13/06/2016 | ≥10 °C GDD | 87 (6) | 945 (8) | 854 (9) |

Although variations existed in the planting dates in these multi-location trials, the results showed small changes in the duration of the individual stage during the development of the crop. In other words, during emergence to flowering and flowering to maturity, both maize varieties PHB30-G19 and Melkasa-4 of ≥8 °C accumulated temperature and GDD requirements which were not different in their values between sites while considering small variations between sowing dates. Such outcomes support the logic that implies that photoperiod has little influence on the maize developmental stages in the selected tropical regions.

Results in Table 3 revealed that an increase in mean temperature value leads to a decrease in the total length of the vegetative and reproductive phases of all maize varieties tested in Ethiopia; a similar tendency was observed in Tanzania (Table 4). On the contrary, in Nigeria, where the mean temperature was almost identical in selected sites, there was no significant difference in the length of the growing period for all selected maize varieties.

**Table 3.** Summary of the number of days from sowing to emergence, emergence to flowering, and flowering to maturity, respectively, for BH660 type hybrid maize variety for 3 planting dates in 5 locations of Ethiopia.

| Locations | Planting Dates | Altitude (m) | Average Min. Temperature (°C) | Average Max. Temperature (°C) | Days from Sowing to Emergence | Days from Emergence to Flowering | Days from Flowering to Maturity |
|---|---|---|---|---|---|---|---|
| DEDESSA | 23/05/2016 | 1224.82 | 18.50 | 32.21 | 7 | 65 | 66 |
| DEDESSA | 30/05/2016 | 1231.44 | 18.55 | 32.51 | 7 | 68 | 69 |
| DEDESSA | 06/06/2016 | 1237.00 | 18.70 | 32.74 | 7 | 71 | 70 |
| BAKO | 25/05/2016 | 1640.30 | 14.11 | 27.26 | 8 | 80 | 80 |
| BAKO | 03/06/2016 | 1648.89 | 14.23 | 27.26 | 7 | 81 | 80 |
| BAKO | 12/06/2016 | 1647.79 | 14.67 | 27.26 | 7 | 83 | 85 |
| HOLLETA | 13/06/2016 | 2351.59 | 08.79 | 22.40 | 9 | 112 | 118 |
| HOLLETA | 20/06/2016 | 2369.03 | 08.80 | 22.40 | 9 | 103 | 101 |
| HOLLETA | 27/06/2016 | 2369.03 | 08.80 | 22.40 | 10 | 122 | 119 |

**Table 4.** Summary of the number of days from sowing to emergence, emergence to flowering and flowering to maturity respectively; for MERU HB 513 type hybrid maize variety for 3 planting dates in 3 locations of Tanzania.

| Locations | Planting Dates | Altitude (m) | Average min. Temperature (°C) | Average max. Temperature (°C) | Days from Sowing to Emergence | Days from Emergence to Flowering | Days from Flowering to Maturity |
|---|---|---|---|---|---|---|---|
| MIWALENI | 16/05/2016 | 1581.30 | 20.00 | 31.50 | 7 | 66 | 65 |
| MIWALENI | 23/05/2016 | 1581.30 | 20.00 | 31.50 | 7 | 68 | 69 |
| MIWALENI | 30/05/2016 | 1554.50 | 20.00 | 31.50 | 7 | 67 | 70 |
| UYOLE | 23/12/2015 | 1769.26 | 12.69 | 25.42 | 9 | 82 | 77 |
| UYOLE | 30/12/2015 | 1769.26 | 12.69 | 25.42 | 8 | 85 | 80 |
| UYOLE | 06/01/2016 | 1769.26 | 12.69 | 25.42 | 8 | 80 | 75 |
| IGERI | 31/12/2015 | 2212.73 | 11.36 | 20.28 | 9 | 103 | 108 |
| IGERI | 06/01/2016 | 2212.73 | 11.36 | 20.28 | 9 | 109 | 102 |
| IGERI | 13/01/2016 | 2212.73 | 11.36 | 20.28 | 9 | 106 | 93 |

## 4. Discussion

The cameras on the mobile smartphones allowed the capturing of images of plant leaves with a pest infestation or signs of damage by disease. Such features allow the system for data collection presented here to serve for pest and disease detection and diagnosis. This facilitated diagnosis by plant pathologists using the images for analyses, which immensely contributed to disease detection and triggered the need for rapid intervention.

The data collection approach presented in this paper provides key insights for improving the integrity of field data collection in agricultural research [32]. This transcends ordinary usage of these gadgets as simple communication devices but involves harnessing the mobile smartphone technologies that support the collection of different data types (text, logical, numerical, photos), which users can then upload, transfer, store, and access via the cloud-based server [1,11,32]. The overall system provides real-time supervision, which is a unique advancement over previous methods of conducting agronomic field data collection [32,33]. For example, there have been several reported issues of data fabrication when field surveys are conducted with paper and pen [34]. Therefore, web-based technologies such as the ODK approach presented in this study provide a convenient means to perform regular checks that enhance the detection of data fabrication [34]. The global positioning system (GPS) capabilities further enabled the tracking of the accuracy of the locations where data were collected based on the initially generated, geo-referenced sample points. Because the forms that we used in this study captured the start and end time, as well as the total time that was taken to complete each survey, the ODK offers a unique and real-time monitoring system. Assuredly, the electronic graphs and instantaneous information provided means of quality control and further completely reduced the cost of data entry.

The advent of mobile smartphones and particularly the entry of affordable devices into the mobile phone market further opened opportunities for the collection of new, complex, and unique data

types such as the maize phenology that is presented here [1,11,26]. It is noteworthy that agricultural research and intervention projects can greatly benefit from the versatility of mobile smartphones for capturing precise locational information, high-resolution photographs, and encrypting information in the barcoded format [1]. These new data types can then be integrated with traditional data types to improve the analysis and interpretation of the data and facilitate knowledge discovery. It is anticipated that major technological trends in the 21st century will continue to favor the proliferation and affordability of mobile smartphones in remote areas and among poor farmers. For instance, the increased shift of mobile technologies from closed to open operating systems will likely drive down the cost of the software segment of the mobile smartphone devices. Further, advances in cloud-based storage, which has enhanced flexibility in computing and storage space, will likely reduce processing power requirements and drive down the cost per unit device [5].

Despite the advantages that mobile smartphones bring, the major challenge for their usage in some rural and low-income areas within the selected countries is the loss of battery power. This is a valid concern in most remote rural locations with no electrical power supply. Solar-powered chargers and power banks are a potential solution and the data collector can carry several mobile smartphones to alternate while in the field in case the charge of the battery is exhausted.

The growing degree days (GDD) is a popular temperature index employed to estimate plant development, and this index is often used to guide farmers in management decision-making such as timing of irrigation and pesticide application [30,35]. However, in the context of this study, the GDD was calculated to evaluate the level of influence of photoperiod or day length during the developmental phases of the selected maize varieties in chosen countries [35]. The values of GDD for each phase of development obtained from different planting dates were comparatively constant for each maize variety, as published by other studies [31,35]. Variations in sowing depth and seedbed surroundings, which affect soil temperature, soil water content, surface residue, and soil type, may account for the small discrepancies observed during the sowing to emergence phases. Another possible explanation is that traits selected by breeders in the regions cause the varieties to become insensitive to photoperiod [30,35]. Moreover, the countries selected for the study are located near the equator, where the variation in photoperiod is also very small.

## 5. Conclusions

Conducting large-scale agronomy data collections requires real-time quality control and enumerators' supervision. This was achieved using mobile smartphones, which is an easier and more time-efficient option compared to the paper and pen-based method. Such an approach has the prospect of being scaled up in various ways for groups and studies of almost any kind and size. The target of enhancing the productivity of smallholder farmers, especially in maize-based systems, is achievable given the development of good data collection using emerging methods that can generate accurate information and support big data analytics for decision support. We anticipate that such a revolution in data collection has the potential to reshape the usage and application of agronomic information. Within the next decade, the adoption of this emerging innovative approach for collecting data by research institutions will likely enhance the pathway towards open access and big data archiving for sustainable development.

**Supplementary Materials:** The following are available online at http://www.mdpi.com/2073-4395/10/9/1363/s1, Table S1: Hybrid and open-pollinated varieties (OPV) of maize with their traits and characteristics obtained from Ethiopia and planted in Dedessa, Uke, Bako, Ambo, Holleta, and Melkassa respectively, Table S2: Hybrid and open-pollinated varieties (OPV) of maize with their traits and characteristics obtained from Nigeria and planted in Lere, Bayero University Kano (BUK), Samaru, Dambatta, Illorin, and Talata respectively, Table S3: Hybrid and open-pollinated varieties (OPV) of maize with their traits and characteristics obtained from Tanzania and planted in Igeri, Mitalula, Seatondale, Suluti, Oyole, Sari, Monduli juu, Miwaleni, Minja and Siha respectively.

**Author Contributions:** Conceptualization, H.E.Z.T., T.B., K.F.M., I.M. and P.C.; methodology, H.E.Z.T.; software, H.E.Z.T.; validation, H.E.Z.T., T.B., K.F.M., I.M. and A.A.A.; formal analysis, H.E.Z.T. and A.A.A.; investigation, H.E.Z.T., T.B., K.F.M. and I.M. resources, P.C. and B.V.; data curation, H.E.Z.T., T.B., K.F.M. and J.A.;

writing—original draft preparation, H.E.Z.T., B.T.M., T.B. and J.A.; writing—review and editing, B.V., H.E.Z.T., P.C. and B.T.M.; visualization, B.T.M.; supervision, P.C. and B.V.; project administration, P.C.; funding acquisition, P.C. and B.V. All authors have read and agreed to the published version of the manuscript.

**Funding:** The present study was executed by the International Maize and Wheat Improvement Centre (CIMMYT) and International Institute of Tropical Agriculture (IITA) as part of the TAMASA (Taking Maize Agronomy to Scale in Africa) project, made possible by the generous support of the Bill and Melinda Gates Foundation (contract OPP1113374). Any opinions, findings, conclusions, or recommendations expressed in this publication are those of the authors and do not necessarily reflect the view of the donor.

**Conflicts of Interest:** The authors declare no conflict of interest.

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
