# Peer review of "Rapid Acquisition, Management, and Analysis of Spatial Maize (Zea mays L.) Phenological Data—Towards ‘Big Data’ for Agronomy Transformation in Africa"

_agronomy, doi:10.3390/agronomy10091363_

Round 1

Reviewer 1 Report

Nicely presented paper, Some minor editing suggestions:

Line 88: add "the" between in and case

Line 90: maize plant change to maize plants

Line 129: add program after training

Line 138: Replace Referring with something else perhaps "Due"

Line 185: Equation needs to be written using Equation editor

Line 200: image A is out of focus

Lines 230 -236 add nothing as the Tables 1 -3 say the same information much more clearly.

Line 280: add either "a" or "the" between via and cloud.

Line 304: add bring after smartphones

Line 391: reference begins with and

Author Response

Line 88: We have added “the” as suggested

Line 90: This was done as suggested

Line 138: Program’ was added after training

Line 138: We rephrased the sentence it now reads: “The features for collection, transfer, and archiving of geospatially referenced data of the ODK app were used”
Thank you for the suggestion.

Line 185: The equation was reconstructed using the equation editor

Line 200: This was done as suggested

Lines 230 -236: Thank you for that observation we have removed them from the text as suggested

Line 304: The word “the” was added. Thank you

Line 391: This reference was corrected. Thank you for the observation

Reviewer 2 Report

Overall: Exciting use of existing technology to help fill a critical need. One thing I don’t see is any kind of IRB approval for use with human subjects. I realize that you are not directly surveying people, but in line 157 “removal of sensitive information” is specifically noted. I imagine a name or farm identifier was used during data collection was used during this process. Also, more presentation of results surrounding the actual use of smartphones and the platform used to collect the data is needed. As it stands, your results and discussion do not match.

Line 59: What does ICT stand for? Is used throughout and has not been explained.

Line 66: Mobile phones smartphones, mobile smart-phones, and smart mobile phones have all been used in close proximity. Suggest sticking with one (preferably mobile smart-phones) throughout the text.

Line 169: So fields were a convenience sample based on whether or not weather stations were available in the area? Were any other data collected such as soil quality, previous yield data, etc., to account for variability within each country and even each section of the country?

Table 1: Suggest increasing size or decreasing text so accumulated temperature is not spread across 4 lines.

Results/Discussion: You have focused on maize data during the results section, but not information on the data collection capabilities you allude to in the discussion section. You make several claims that using smartphones improved data accuracy, reduced data fabrication, and could be monitored through the ODK platform. How exactly could that be accomplished? And why was that information not presented in the results section? The results section needs to be amended to match the main focus of the discussion - the ability to effectively collect data using a smartphone platform.

Author Response

Line 59: ICT stands for “Information, Communication and Technology”. We have since defined it in the text. Thank you

Line 66: Thank you for this critical observation. In the revised manuscript we have consistently used the word mobile smart-phone(s)

Line 160: Thank you for this comment. We collected the soil data and yield, however, the main objective of this study was to evaluate the temperature needs of the different varieties as well as to test the use of the ODK and farmers input for such a complex phenomenon.

Table 1: We decreased the text and the words accumulated temperature no longer spread across 4 lines.

Results/Discussion: However the results we present in this study show the capability of the ODK platform by using a demonstration of the plot data collected by smallholder farmers. In the results section, we allude to the type and quality of the data that was obtained. As such we, therefore, analyse that data within the demonstration application of the data which was to evaluate the maize phenology, whose flow was followed in the discussion section.

We have also commented in lines 288-293 that The global positioning system (GPS) capabilities further enabled tracking the accuracy of the locations where data were collected based on the initial generated geo-referenced sample points. Additionally, because the captured the start and end time as well as the total time taken to complete each survey, it facilitated the easy of checking the accuracy, fabrication

Reviewer 3 Report

Data collection and application workflow is briefly explained. It is necessary to describe the related explanation in the form of a schematic diagram so that the reader can understand it easily.

Author Response

Thank you for this comment, we have provided Figure 2 schematic diagram which graphically describes the workflow that was used in this study. Thank you very much